# Comparative Analysis of Reconstitution and Solubility of Two Poly-L-Lactic Acid Fillers for Medical Applications

**DOI:** 10.3390/polym17131778

**Published:** 2025-06-27

**Authors:** Pawel Kubik, Wojciech Gruszczyński, Monika Filipowska

**Affiliations:** K-LAB Badania i Rozwój, 81-312 Gdynia, Poland; pawel.kubik@k-lab.com.pl (P.K.);

**Keywords:** poly L-lactic acid filler, tissue integration, biocompatibility, regenerative aesthetics, outcomes

## Abstract

Objective: This study aimed to evaluate the reconstitution and solubility characteristics of two injectable poly L-lactic acid (PLLA) formulations, the next-generation injectable PLLA-LASYNPRO™ and a previous-generation PLLA (PLLA-SCA), through in vitro analysis. Methods: The study, conducted from 20 November to 11 December 2024, involved reconstituting PLLA-LASYNPRO™ in 5 mL saline solution and PLLA-SCA in 8 mL sterile water or saline (as specified in the instructions for use). Reconstitution processes were recorded using a 4K camera, and microscopic and sieve-based analyses were performed at various time points (1–120 min). The study used standardized methods for reconstitution, observation, and sample collection, including a light microscope and micrometric sieves. Results: PLLA-LASYNPRO™ exhibited rapid, uniform reconstitution, resulting in a homogeneous milky-white solution with no foam formation. In contrast, PLLA-SCA showed a two-phase mixture with foam formation, requiring more vigorous shaking. Over time, PLLA-LASYNPRO™ displayed gradual sedimentation, with no foam, while PLLA-SCA showed a progressive increase in foam volume and sedimentation. Microscopic analysis revealed consistent particle morphology, with PLLA-LASYNPRO™ particles being smaller and homogeneous, and PLLA-SCA exhibiting larger, irregular particles prone to aggregation. Sieve analysis showed that PLLA-LASYNPRO™ produced minimal sediment, whereas PLLA-SCA consistently produced sediment on both 71 µm and 200 µm sieves. Conclusions: PLLA-LASYNPRO™ and PLLA-SCA display distinct reconstitution behaviors, with PLLA-LASYNPRO™ showing more consistent characteristics and fewer issues with foam and sedimentation. These differences may impact clinical applications and require consideration in treatment planning.

## 1. Introduction

Injectable fillers have gained widespread recognition as effective, minimally invasive alternatives to surgical procedures for facial enhancement and rejuvenation [1,2].

Poly L-lactic acid (PLLA), an emerging biodegradable polymer, has been introduced as an alternative filler that promotes collagen synthesis while supporting fibrous tissue regeneration at the injection site [3,4,5,6]. Unlike hyaluronic acid (HA) fillers, which primarily provide immediate volumization, PLLA elicits a localized stimulatory response that enhances skin texture and volume over time, with effects lasting up to two years [5,6,7,8,9]. The commonly used fragmented PLLA formulations are suspended in sterile water before injection, requiring a latency period before visible results manifest. Despite its biocompatibility, adverse reactions, including significant inflammatory responses, have been reported [10,11].

The biocompatibility of PLLA and its ability to stimulate collagen synthesis are influenced by various factors, including material properties, patient characteristics, and injection technique [9,12]. These elements collectively shape the host response and are crucial in ensuring a controlled and predictable therapeutic outcome. While the precise molecular mechanisms underlying PLLA-induced collagen production remain incompletely understood, research has examined the biochemical pathways activated in fibroblasts, adipocytes, and macrophages in response to PLLA implantation [5,6,7,8,9,12]. Differences in injectable PLLA formulations, particularly in their integration into tissue and degradation kinetics, are also of significant interest [13,14].

Researchers are continually refining PLLA formulations to optimize results and minimize potential side effects. Potential advances may include modifying particle size and shape, concentration, or delivery mechanism to enhance tissue integration and longevity. This study aimed to evaluate the reconstitution and solubility characteristics of the next-generation injectable PLLA-LASYNPRO™ (Juläine™, Nordberg Medical AB, Stockholm, Sweden; LoviSelle^TM^, Changchun Sinobiom Co., Ltd., Changchun, China) and how these compare with those of a first-generation PLLA filler (Sculptra^®^, Galderma Sweden, Uppsala, Sweden) (PLLA-SCA).

## 2. Methods

The reconstitution and solubility characteristics of two distinct PLLA fillers, the next-generation injectable PLLA-LASYNPRO™ and PLLA-SCA, were evaluated through in vitro analysis.

The study was conducted between 20 November 2024 and 11 December 2024. The initial phase involved the preparation of materials and the validation of the research methodology.

To ensure consistency throughout the study, standardized procedures and conditions were implemented for reconstitution and observation of the products. During this phase, the most suitable research techniques were selected to meet the objectives of the study, including the following:Capturing reconstitution processes on a shadowless table using a 4K camera (MX Brio^TM^, Logitech, 1015 Lausanne, Switzerland).Conducting analyses with a digital light microscope Levenhuk D95L LCD (Levenhuk, Inc., Tampa, FL, USA), which involved digital recording of both static and dynamic images using an 8 MP built-in camera,Utilizing certified micrometric sieves with hole diameters of 200 μm and 71 μm, coupled with digital image recording via a 12 MP camera.Following the validation process, the main phase of the study was carried out, culminating in the reporting of the findings.

Given the nature of the research, obtaining approval from a Bioethics Committee or any additional consents or permits was not required.

## 3. Reconstitution Procedures

### 3.1. Reconstitution

PLLA-LASYNPRO™

PLLA-LASYNPRO™ was reconstituted using 5 mL of sterile physiological balanced saline solution for injection, in accordance with the Instructions for Use (IFU) [15]. The procedure involved injecting the saline into the vial and shaking vigorously for approximately 1 min to ensure complete reconstitution.

PLLA-SCA

PLLA-SCA was reconstituted using 8 mL of sterile water for injection, with an additional 5 mL of water for injection used for the observation time point of 120 min. The IFU for PLLA-SCA specifies that 8 mL of water is required for reconstitution if the product is used within 120 min after preparation [16]. However, for longer observation times, 5 mL of water is sufficient, provided that the preparation is allowed to stand for at least 120 min before administration.

The reconstitution procedures for PLLA-SCA were as follows:For 8 mL reconstitution: 5 mL of water for injection was added to the vial, followed by vigorous shaking for 1 min. Then, an additional 3 mL of water was added, and the vial was shaken for a further 1 min.For 5 mL reconstitution: 5 mL of water for injection was added to the vial, and the vial was shaken vigorously for 1 min.

The reconstitution fluids were maintained at room temperature (22 °C ± 1.5 °C), and all injections were performed using sterile 18G needles.

### 3.2. Observation Conditions

The observation of the reconstituted products was conducted under standardized environmental conditions with a temperature of 21 °C ± 2 °C and humidity of 50% ± 5%.

The reconstituted products were allowed to stand for the following observation time points: 1, 2, 5, 10, 20, 40, 60, 90, and 120 min. At each time point, the products were shaken vigorously (as per the IFU) before collecting the material for further analysis.

### 3.3. Sample Collection

At each observation time point, material was collected as follows:PLLA-LASYNPRO™: As the product is homogeneous, no separated layers were collected.PLLA-SCA: Due to its inhomogeneity and the clear instructions in the IFU, care was taken to avoid collecting any foam formed during reconstitution.

The following quantities were collected using sterile 18G needles:0.5 mL for microscopic examination.1.5 mL for evaluation on a 200 μm micrometric sieve.1.5 mL for evaluation on a 71 μm micrometric sieve.

### 3.4. Macroscopic Observations

Macroscopic observations were conducted under controlled lighting, temperature, and humidity conditions, as outlined in the general methodology.

#### 3.4.1. Recording Setup

Reconstitution processes were recorded using an ultra-high-definition 4K camera (MX Brio^TM^, Logitech, 1015 Lausanne, Switzerland) mounted above a shadowless table, which was modified with a black, matt base for optimal contrast and particle visibility. Side lighting and overhead illumination from shadowless lamps (total power of 255 W) were used to enhance image clarity.

The reconstitution process was recorded for each product at each time point, from the moment the solvents were injected into the vials until the end of the observation period. A new vial of each product was used for each recording. Additionally, separate recordings were made showing the flow of both test products through the 30G/13 mm and 33G/13 mm needles.

The recorded footage was processed using Adobe Premiere Pro ver.24.6.5 (Adobe Systems Sofware, Ltd., Dublin, Ireland) to add time stamps and encode the video into a suitable file format for further analysis.

#### 3.4.2. Microscopic Observations

Microscopic observations were conducted following the re-shaking procedure as outlined in the product IFUs [15,16]. Microscopic examination was performed using the Levenhuk D95L LCD digital microscope (Levenhuk, Inc., Tampa, FL, USA). Three separate drops of each tested product were placed on a glass slide. Observations were conducted on the drops directly, as this closely mimicked the behavior of the products in tissues. For PLLA-SCA, care was taken to avoid the collection of foam, whereas no such precaution was necessary for PLLA-LASYNPRO™ due to the absence of foam formation.

During the validation phase, it was found that covering the samples with a coverslip distorted the visibility of PLLA formation, which is why the product was observed without a coverslip.

For PLLA-LASYNPRO™, all samples were reconstituted in 5 mL of saline solution [15]. For PLLA-SCA, samples were reconstituted in 8 mL of saline solution (as specified in the IFU) [16] and 5 mL of sterile water for injections. All examinations were conducted using the samples reconstituted in 8 mL, except for the 120 min timepoint, which was also evaluated using the solution reconstituted in 5 mL.

#### 3.4.3. Micrometric Sieve Observations

The solutions were passed through standardized and certified micrometric sieves (ATEST, Poland) with pore sizes of 71 µm and 200 µm (Appendix A).

At each time point, 1 mL of the reconstituted product was filtered through the respective sieve. The sieves were then left to dry for six hours before sediment analysis. To document any residues, photographs were taken, providing insights into the particle size distribution of the products over time.

The presence of white sediment was attributed to the deposition of PLLA particles on the sieves, whereas the observed “wet” trace was primarily associated with the drying process of carboxymethylcellulose on the sieve meshes.

For PLLA-LASYNPRO™, all samples were reconstituted in 5 mL of saline solution, whereas PLLA-SCA samples were reconstituted in 8 mL of saline solution.

## 4. Results

### 4.1. Macroscopic Observations

#### Immediately After Reconstitution

Upon reconstitution, a distinct difference between the two medical devices was observed immediately (Figure 1).

PLLA-LASYNPRO™: Reconstitution occurs rapidly and uniformly. There is no foaming, and the mixture does not adhere to the walls of the vial. The resulting solution is homogeneous, with a milky-white appearance (Figure 1A).

PLLA-SCA: Reconstitution requires more intensive shaking. Immediately after mixing, a two-phase mixture forms, consisting of a milky-white solution and white foam. Additionally, there is significant settling of precipitated white particles on the vial walls. This phenomenon occurs regardless of whether 5 mL or 8 mL of water is used for reconstitution (Figure 1B,C, respectively).

Minutes 1–20

PLLA-LASYNPRO™: The solution remained stable with no observable differences in appearance (Figure 2A,D,G,J,M, respectively).

PLLA-SCA: A gradual separation of the liquid and foam phases became evident (Figure 2). This was likely due to the sedimentation of PLLA particles suspended in the liquid solution of carboxymethylcellulose and mannitol. Over time, the transparency of the liquid phase increased, indicating the separation of phases. Additionally, the volume of foam covering the liquid phase increased subtly with time (Figure 2B,C,E,F,H,I,K,L,N,O,Q, respectively).

Minutes 40–120

PLLA-LASYNPRO™: At the 40 min time point, the first signs of sedimentation of PLLA particles were observed. This phenomenon progressed gradually, with the transparency of the upper layers of the solution increasing as time passed. Notably, this sedimentation was not accompanied by any foam formation (Figure 2P,S,V,Y, respectively).

PLLA-SCA: At the subsequent time points, sedimentation continued progressively. The transparency of the liquid phase continued to increase, while the foam volume increased slightly, maintaining a stable presence of precipitated structures on the vial walls. These observations remained consistent over time (Figure 2Q,R,T,U,W,X,Z,AA, respectively).

### 4.2. Microscopic Observations

No significant differences were observed in the microscopic images of identical solutions at different time points between PLLA-SCA and PLLA-LASYNPRO™. Throughout the study, both fillers exhibited consistent characteristics. However, despite the apparent similarity in their composition, they demonstrated notable differences in particle morphology and behavior.

#### 4.2.1. PLLA-LASYNPRO™ Microscopic Characteristics

The microscopic analysis of PLLA-LASYNPRO™ revealed a solution consisting of multiple particles, averaging 40–60 particles per field of view at 400× magnification. These particles were generally small, round, and homogeneous in shape, with sizes ranging from 15 to 50 μm. Occasionally, small, trapped air bubbles (up to approximately 80 μm in size) were observed between the particles. Notably, the PLLA particles did not show any tendency to aggregate, even when they were in direct contact with one another or with the air bubbles. This behavior was consistent across all observation time points (Figure 3).

#### 4.2.2. PLLA-SCA Microscopic Characteristics

The PLLA-SCA solution exhibited a lower particle density, averaging 6–15 particles per field of view at 400× magnification. The particles were irregular in shape, characterized by sharp edges and a size range of approximately 15 to 120 µm. Additionally, small air bubbles, similar to those observed in the PLLA-LASYNPRO™ solution, were present, measuring up to approximately 80 µm. However, the most notable distinction between PLLA-SCA and PLLA-LASYNPRO™ was the pronounced tendency of PLLA particles in the PLLA-SCA solution to aggregate (Figure 3).

### 4.3. Micrometric Sieve Observations

PLLA-LASYNPRO™

A visible sediment trace was detected only on the 71 µm sieve at the 1 min mark. No sediment was observed in any subsequent measurements taken between 2 and 120 min (Figure 4, Appendix A).

No sediment was observed on the 200 µm sieve at any of the tested time points.

PLLA-SCA

White sediment was consistently present on both the 71 µm and 200 µm sieves, irrespective of reconstitution time. At the 120 min time point, the sediment observed on the 200 µm sieve appeared slightly larger and thicker in the 5 mL reconstituted solution compared to the 8 mL reconstituted solution (Figure 4, Appendix A).

## 5. Discussion

The results of this study revealed substantial differences in the physicochemical properties and behavior of PLLA-LASYNPRO™ and PLLA-SCA formulations, despite their similar composition. Notably, PLLA-LASYNPRO™ exhibited rapid, uniform reconstitution resulting in a homogenous solution, whereas PLLA-SCA exhibited a two-phase mixture with foam formation and a greater tendency for particle aggregation and sedimentation.

Macroscopic Observations

Macroscopic analyses revealed notable differences between PLLA-LASYNPRO™ and PLLA-SCA filler formulations. PLLA-LASYNPRO™ formed a homogeneous solution with small, smooth, round PLLA particles that remained well-suspended, sedimented slowly, and exhibited no foam formation or visible deposits on micrometric sieves. In contrast, PLLA-SCA contained larger, irregularly shaped PLLA particles that sedimented rapidly, generated significant foam—raising clinical concerns as foam should not be injected, per Sculptra’s instructions for use [16].

Particle morphology and size both influence the biostimulatory and inflammatory characteristics of injected collagen stimulators [17]. It has previously been reported that the particle size of a filler should be sufficiently large (above 20 μm) to avoid phagocytosis, yet small enough (below 100 μm) to easily pass through a needle [14]. Increasing size and surface irregularity have been shown to induce inflammatory macrophage activation and can increase the release of proinflammatory cytokines [18,19,20]. Smaller, more homogeneous particles tend to elicit a more moderate inflammatory response and promote progressive tissue repair [9,21,22,23,24,25]. The behavior of the product may also be influenced by the concentration of the CMC carrier. The PLLA-LASYNPRO™ formulation contains 50% less CMC than PLLA-SCA [15,16]. CMC is a known viscosity modifier and can influence the physicochemical properties of polymeric systems, potentially affecting particle behavior and aggregation [26,27,28]. Homogeneous dispersion of particles in the CMC carrier is essential for preventing the formation of non-inflammatory nodules of accumulated product.

The compositional difference may contribute to the distinct macroscopic behaviors observed between the two products. Interestingly, the manufacturer’s guidelines for PLLA-SCA [16] recommended a minimum reconstitution time of 120 min for more concentrated solutions; however, macroscopic analysis revealed no significant differences between solutions prepared with 5 mL or 8 mL of solvent, nor did reconstitution time appear to influence the overall macroscopic behavior of the product.

Microscopic Observations

Microscopic analyses further elucidated the disparities observed macroscopically between PLLA-LASYNPRO™ and PLLA-SCA filler formulations. PLLA-LASYNPRO™ exhibited well-dispersed PLLA particles without signs of aggregation, suggesting a formulation conducive to gradual and sustained tissue integration. Conversely, PLLA-SCA particles displayed a pronounced propensity to aggregate, forming structures exceeding 1000 µm, particularly in the presence of air bubbles. This aggregation aligns with findings from previous studies, which have documented similar phenomena and associated them with adverse tissue responses, including acute inflammation and fibrosis [22,29,30].

Additionally, this investigation identified morphological differences in the shape of the particles between the two products. PLLA-LASYNPRO™ microspheres were smooth, homogeneous, and spherical, with diameters ranging from 15 to 50 µm. In contrast, PLLA-SCA microflakes consisted of irregularly shaped particles with dimensions ranging from 15 to 120 µm along their major axis, corroborating previous reports on the irregularity of PLLA-SCA particles [14,31].

Light microscope observations on circularity and roundness further emphasized these differences. PLLA-LASYNPRO™ microspheres exhibited greater circularity and roundness than PLLA-SCA particles. The tightly clustered circularity measurements for PLLA-LASYNPRO™ indicated a more uniform and symmetrical shape, while PLLA-SCA particles showed a wider distribution, reflecting greater irregularity and heterogeneity in particle structure.

Particle morphology plays a crucial role in material–immune interactions, with features such as circularity, roundness, and aspect ratio influencing inflammasome activation [19]. The body recognizes shape and size discrepancies in foreign materials, modulating the immune response. Smooth, spherical particles generally provoke less inflammatory reaction, while spikier particles can disrupt phagosomal membranes, leading to increased inflammation [20,32,33].

Micrometric Sieve Observations

Micrometric sieve observations further reinforced the significant differences between PLLA-LASYNPRO™ and PLLA-SCA in terms of particle aggregation and sedimentation behavior. PLLA-LASYNPRO™, which exhibited well-dispersed, non-aggregating PLLA particles, did not leave detectable sediment on either the 71 µm or 200 µm sieves. On the contrary, PLLA-SCA consistently produced visible white sediment on both sieve sizes, with a concentration-dependent effect observed on the 200 µm sieve. These findings suggest that PLLA-SCA particles tend to aggregate more strongly, potentially influencing its clinical performance [24,25]. This is consistent with the literature highlighting the importance of the solubility and physicochemical properties of PLLA, which affect its behavior both in aqueous solutions and in tissues, impacting its integration and inflammatory response, and overall safety [24,25,34,35,36].

## 6. Conclusions

In conclusion, this study provided a comprehensive comparison of the morphological characteristics and behavior of PLLA-LASYNPRO™ and PLLA-SCA fillers. Notable differences in particle shape, circularity, roundness, and aspect ratio were observed between the two formulations. PLLA-LASYNPRO™ consisted of homogeneous, smooth, and spherical microspheres, while PLLA-SCA contained irregularly shaped microflakes. These morphological disparities may have significant implications for both aesthetic and safety outcomes. The irregularly shaped particles in PLLA-SCA could potentially elicit a more pronounced inflammatory response, whereas the uniform, spherical particles in PLLA-LASYNPRO™ may contribute to a reduced inflammatory reaction. The distinct physicochemical properties observed in these commercially available PLLA products are likely to influence the biological responses seen in clinical practice and should be considered when selecting an appropriate PLLA-based treatment for aesthetic procedures. However, further clinical studies evaluating the impact of PLLA-LASYNPRO™ and PLLA-SCA fillers on aesthetic outcomes and safety are necessary to fully assess the clinical relevance of these findings in conjunction with the existing body of literature.

## Figures and Tables

**Figure 1 polymers-17-01778-f001:**
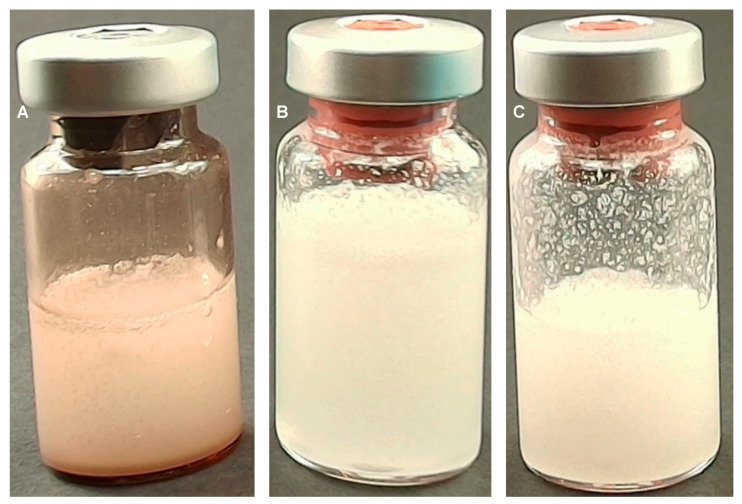
Images taken immediately after reconstitution of the PLLA-LASYNPRO™ (**A**); PLLA-SCA reconstituted in 8 mL (**B**); and PLLA-SCA reconstituted in 5 mL (**C**).

**Figure 2 polymers-17-01778-f002:**
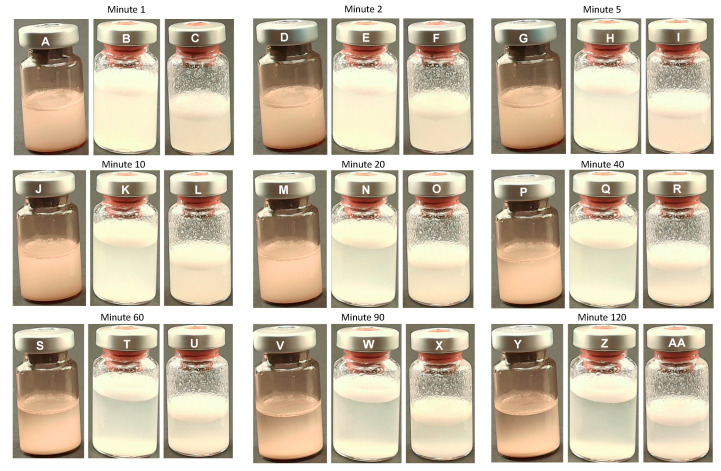
Overview of the macroscopic observations in the PLLA-LASYNPRO™ and PLLA-SCA reconstituted using 8 mL or 5 mL at different time points. PLLA-LASYNPRO™: Minute 1 (**A**); Minute 2 (**D**); Minute 5 (**G**); Minute 10 (**J**); Minute 20 (**M**); Minute 40 (**P**); Minute 60 (**S**); Minute 90 (**V**); Minute 120 (**Y**). PLLA-SCA 8 mL: Minute 1 (**B**); Minute 2 (**E**); Minute 5 (**H**); Minute 10 (**K**); Minute 20 (**N**); Minute 40 (**Q**); Minute 60 (**T**); Minute 90 (**W**); Minute 120 (**Z**). PLLA-SCA 5 mL: Minute 1 (**C**); Minute 2 (**F**); Minute 5 (**I**); Minute 10 (**L**); Minute 20 (**O**); Minute 40 (**R**); Minute 60 (**U**); Minute 90 (**X**); Minute 120 (**AA**).

**Figure 3 polymers-17-01778-f003:**
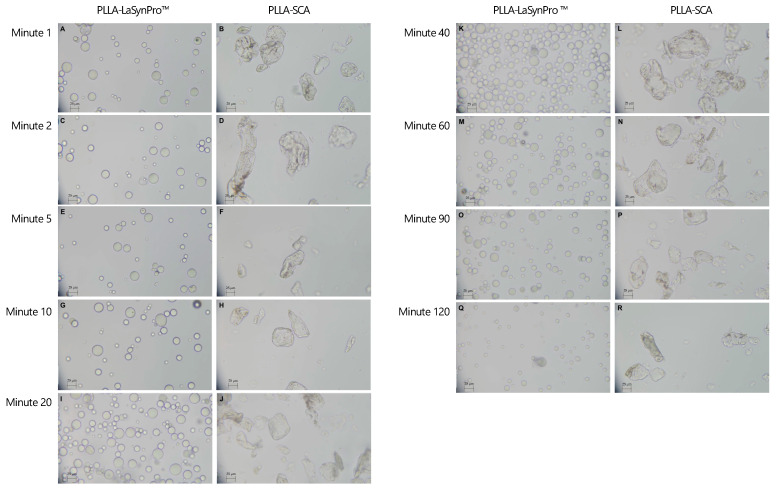
Overview of the microscope droplet–smear observations with 400× magnification in the PLLA-LASYNPRO™ and PLLA-SCA ions at different time points. PLLA-LASYNPRO™: Minute 1 (**A**); Minute 2 (**C**); Minute 5 (**E**); Minute 10 (**G**); Minute 20 (**I**); Minute 40 (**K**); Minute 60 (**M**); Minute 90 (**O**); Minute 120 (**Q**). PLLA-SCA: Minute 1 (**B**); Minute 2 (**D**); Minute 5 (**F**); Minute 10 (**H**); Minute 20 (**J**); Minute 40 (**L**); Minute 60 (**N**); Minute 90 (**P**); Minute 120 (**R**). Scale bars (lower left corners) set to 25 µm.

**Figure 4 polymers-17-01778-f004:**
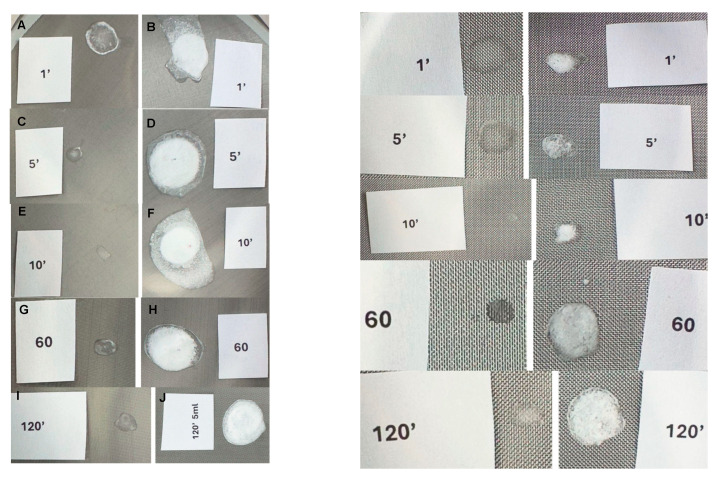
Sediment trace observed on the 71 µm sieve with the PLLA-LASYNPRO™ and PLLA-SCA (8 mL) solutions at different time points. PLLA-LASYNPRO™: (**A**,**C**,**E**,**G**,**I**) slides. PLLA-SCA: (**B**,**D**,**F**,**H**,**J**) slides.

## Data Availability

The original contributions presented in this study are included in the article/Appendix A. Further inquiries can be directed to the corresponding author(s).

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
