# Peer review of "Comparative Analysis of Reconstitution and Solubility of Two Poly-L-Lactic Acid Fillers for Medical Applications"

_polymers, 2025, doi:10.3390/polym17131778_

Round 1

Reviewer 1 Report

Comments and Suggestions for Authors

The manuscript entitled "Comparative Analysis of Reconstitution and Solubility of Two Poly-L-Lactic Acid Fillers for Medical Applications", from the authors Pawel Kubik, Wojciech Gruszczyński and Monika Filipowska.

The manuscript is good and contains useful and practical results.

However, the authors themselves did not declare the type of manuscript, which is otherwise required according to the polymers-template.dot file. This manuscript cannot be classified as a scientific paper because scientific research requires a cause-and-effect relationship, i.e. the change of some parameters at the beginning of the experiment should be followed by the change of the obtained results. Thus, this manuscript could be classified as a professional paper, if the journal "Polymers" accepts such manuscripts for publication.

In this manuscript, the properties of two commercial products are discussed. Given that there is a whole series of similar products, it remains unclear why exactly these two products became the subject of analysis? Also, what can be embarrassing is the analysis of two products that have probably received some licenses for human use and are undergoing proper quality control. Inconveniences may arise from the manufacturer who may raise the question of favoring one manufacturer's product over another. On the other hand, if these types of products are not controlled by independent researchers and analysts, manufacturers may feel that they can lower the quality of their products without consequence. That is why it is difficult for me to take a position on the publication of such a manuscript.

The authors did not cite the references according to the polymers-template.dot file. I ask the authors to make references according to the guidelines defined for the journal "Polymers".

Anyway, for now I think that the manuscript should be published in the journal "Polymers" after minor corrections. The final decision is still up to the editor of the journal "Polymers", who must decide whether to publish this type of manuscript.

Comments on the Quality of English Language

I am not qualified enough to assess the quality of the English language.

Author Response

Thank you very much for taking the time to review this manuscript. Please find the detailed responses below and the corresponding revisions.

Comments 1: However, the authors themselves did not declare the type of manuscript, which is otherwise required according to the polymers-template.dot file. This manuscript cannot be classified as a scientific paper because scientific research requires a cause-and-effect relationship, i.e. the change of some parameters at the beginning of the experiment should be followed by the change of the obtained results. Thus, this manuscript could be classified as a professional paper, if the journal "Polymers" accepts such manuscripts for publication.

Response 1: The article type suggested by the authors is ‘research manuscript’.

Comments 2: In this manuscript, the properties of two commercial products are discussed. Given that there is a whole series of similar products, it remains unclear why exactly these two products became the subject of analysis? Also, what can be embarrassing is the analysis of two products that have probably received some licenses for human use and are undergoing proper quality control. Inconveniences may arise from the manufacturer who may raise the question of favoring one manufacturer's product over another. On the other hand, if these types of products are not controlled by independent researchers and analysts, manufacturers may feel that they can lower the quality of their products without consequence. That is why it is difficult for me to take a position on the publication of such a manuscript.

Response 2: The authors thank the reviewer for their comment. Research is continually ongoing to refine PLLA formulations to optimize results and minimize potential side effects. Potential advances may include modifying particle size and shape, concentration, or delivery mechanism to enhance tissue integration and longevity. To this aim, the current study has compared the reconstitution and solubility characteristics of a new PLLA filler with an established market representative.

Comments 3: The authors did not cite the references according to the polymers-template.dot file. I ask the authors to make references according to the guidelines defined for the journal "Polymers".

Response 3: The authors have now styled the references to the format of the journal Polymers.

Reviewer 2 Report

Comments and Suggestions for Authors

Minor comments

Abstract- spaces and proof-reading needed

Introduction - The authors are encouraged to improve upon contextualizing their problem statement and how this research lays foundations to address said challenges.
Results: Figure 2 can benefit from in-figure labels and time-point information to improve readability. Sections 4.2.1/4.2.2 - maintain consistent notation (e.g., um or micrometers). For the observation in Figure 3 the authors are urged to detail the methodlogy, i.e., inspected volume / droplet-smear/ type of substrate etc. The figure is missing a scale bar an din general some formatting for better visual appeal.

Major comments
Discussion: How was  the CMC concentration detected and confirmed? Was it in accorandance to the manufacture's description?

The sentence "micrometer sieve confirm morphology..." - the authors are suggested to minimize broad inferences on shape/ morphology/ orientation and instead better contextualize their observations purely from the physical interaction with a sized-barrier. Further the discussion section needs some references for the suggested physiological response to the product.

Macroscopic observations "smaller, more homogeneous particles tend to elicit a moderate inflammatory response and pro-mote progressive tissue repair (9,17-19)." - here the authors are asked to cite literature and improve the narration on potentially acceptable size- range or values that promote favourable responses and then comment how well their tested materials performed.

"Interestingly, the manufacturer's guidelines for PLLA-SCA (16) recommended a minimum reconstitution time of 120 minutes for more concentrated solutions; however, macroscopic analysis revealed no significant differences between solutions prepared with 5 mL or 8 mL of solvent, nor did reconstitution time appear to influence the overall macroscopic behavior of the product." - And why is that, what comments would the authors like to hypothesize?

"This aggregation aligns with findings from previous studies, which have documented similar phenomena and associated them with adverse tissue responses, including acute inflammation and fibrosis (18,25,26)." - In this case, why did the authors then chose to investigate the same ?

The authors state that "Circularity and roundness analysis", what software did they use to ascertain the 'analysis'. I assume that this is a general comment on observation on the micrographs. The authors are encouraged to improve the narration. Again, what measurement are is being referred to "The tightly clustered circularity measurements for PLLA-LA-SYNPRO™ indicated a more uniform and symmetrical shape," 

I would highly encourage the authors to try to modify the PLLA-SCA to actually understand the interaction or even propose potential strategies to improve the PLLA-SCA such that it can perform like it successful counter part. To do this i suggest trying to break the size distribution either by using an ultrasonic device or alternatively tweak the ion concentration of the solvent to shed some light on the actual behaviour and ways to influence the behaviour of this filler. Further testing, Zeta potential measurements recommended to gain insights into solution/suspension stability.

Author Response

Thank you very much for taking the time to review this manuscript. Please find the detailed responses below and the corresponding revisions.

Point-by-point response to Comments and Suggestions for Authors

Comments 1: Abstract- spaces and proof-reading needed
Response 1:  The document has been proof-read by a native English speaker and spaces inserted where necessary.

Comments 2: Introduction – The authors are encouraged to improve upon contextualizing their problem statement and how this research lays foundations to address said challenges.

Response 2: The authors thank the reviewer for their comment. Research is continually ongoing to refine PLLA formulations to optimize results and minimize potential side effects. Potential advances may include modifying particle size and shape, concentration, or delivery mechanism to enhance tissue integration and longevity. To this aim, the current study has compared the reconstitution and solubility characteristics of a new PLLA filler with an established market representative. The above text has been added to the last paragraph of the introduction.

Comments 3: Results: Figure 2 can benefit from in-figure labels and time-point information to improve readability.

Response 3: In-figure labels was added with time-point information.

Comments 4: Sections 4.2.1/4.2.2 - maintain consistent notation (e.g., um or micrometers).

Response 4: The authors have replaced micrometers with μm throughout the document.

Comments 5: For the observation in Figure 3 the authors are urged to detail the methodlogy, i.e., inspected volume / droplet-smear/ type of substrate etc. The figure is missing a scale bar an din general some formatting for better visual appeal.

Response 5: Figure 3 has in-figure labels. The smear was made from a drop with a volume of approximately 30 µm. The smear was covered with a 24 x 24 mm cover slip. There was no scale on the microscope.

Comments 6: Discussion: How was  the CMC concentration detected and confirmed? Was it in accorandance to the manufacture's description?

Response 6: The CMC concentration is in accordance with that outlined in the products’ respective instructions for use: PLLA-LASYNPRO (45 mg) and PLLA-SCA (90 mg).

Comments 7: The sentence "micrometer sieve confirm morphology..." - the authors are suggested to minimize broad inferences on shape/ morphology/ orientation and instead better contextualize their observations purely from the physical interaction with a sized-barrier. Further the discussion section needs some references for the suggested physiological response to the product.

Response 7: The above sentence has been deleted and discussion on findings from sieve analysis restricted to particle aggregation and sedimentation behavior.

Comments 8: "Interestingly, the manufacturer's guidelines for PLLA-SCA (16) recommended a minimum reconstitution time of 120 minutes for more concentrated solutions; however, macroscopic analysis revealed no significant differences between solutions prepared with 5 mL or 8 mL of solvent, nor did reconstitution time appear to influence the overall macroscopic behavior of the product." - And why is that, what comments would the authors like to hypothesize?

Response 8: Based on observations comparing PLLA-SCA dissolved in 5 ml and 8 ml of solvent we believe the PLLA-SCA IFU recommendation for a reconstitution time of 120 minutes when using 5 ml solution merits further discussion as our analysis showed no benefit of such an approach. 

Comments 9: "This aggregation aligns with findings from previous studies, which have documented similar phenomena and associated them with adverse tissue responses, including acute inflammation and fibrosis (18,25,26)." - In this case, why did the authors then chose to investigate the same ?

Response 9: The authors used PLLA-SCA as the comparator in this study because it is an established market representative with over 20 years of data. The sentence in the quote above indicates that the PLLA-SCA particles had a tendency to aggregate forming structures exceeding 1000 μm. This increases the likelihood of adverse events including inflammation and nodule formation. As PLLA-LA-SYNPRO did not aggregate to the same extent, the inference is that it would be less likely to result in similar adverse events.

Comments 10: The authors state that "Circularity and roundness analysis", what software did they use to ascertain the 'analysis'. I assume that this is a general comment on observation on the micrographs. The authors are encouraged to improve the narration. Again, what measurement are is being referred to "The tightly clustered circularity measurements for PLLA-LA-SYNPRO™ indicated a more uniform and symmetrical shape," 

Response 10: The authors confirm that the circularity and roundness analyses were based on microscopic observations as these features are clearly visible using a simple light microscope. The authors have amended the wording in the text to clarify how particle size and shaper were assessed.

Comments 11: I would highly encourage the authors to try to modify the PLLA-SCA to actually understand the interaction or even propose potential strategies to improve the PLLA-SCA such that it can perform like it successful counter part. To do this i suggest trying to break the size distribution either by using an ultrasonic device or alternatively tweak the ion concentration of the solvent to shed some light on the actual behaviour and ways to influence the behaviour of this filler. Further testing, Zeta potential measurements recommended to gain insights into solution/suspension stability.

Response 11: The authors thank the reviewer for this comment and agree this would be an interesting line of research for a future study.

Round 2

Reviewer 2 Report

Comments and Suggestions for Authors

The authors are thanked for their contributions and appreciated for improving the manuscript. Please find some additional feedback on what we deem important.

Comments 5: For the observation in Figure 3 the authors are urged to detail the methodology, i.e., inspected volume / droplet-smear/ type of substrate etc. The figure is missing a scale bar and in general some formatting for better visual appeal.

Response 5: Figure 3 has in-figure labels. The smear was made from a drop with a volume of approximately 30 µm. The smear was covered with a 24 x 24 mm cover slip. There was no scale on the microscope.

Comment: In response to comment 5, The authors are requested to provide a scale bar, this can be done by imaging a known object (a ruler/ a stage micrometre etc.) and determining the magnification.

Comments 6: Discussion: How was  the CMC concentration detected and confirmed? Was it in accordance to the manufacture's description?

Response 6: The CMC concentration is in accordance with that outlined in the products’ respective instructions for use: PLLA-LASYNPRO (45 mg) and PLLA-SCA (90 mg).

Comment: In response to comment 6, the 'how' still remains unexplained.

Comments 8: "Interestingly, the manufacturer's guidelines for PLLA-SCA (16) recommended a minimum reconstitution time of 120 minutes for more concentrated solutions; however, macroscopic analysis revealed no significant differences between solutions prepared with 5 mL or 8 mL of solvent, nor did reconstitution time appear to influence the overall macroscopic behavior of the product." - And why is that, what comments would the authors like to hypothesize?

Response 8: Based on observations comparing PLLA-SCA dissolved in 5 ml and 8 ml of solvent we believe the PLLA-SCA IFU recommendation for a reconstitution time of 120 minutes when using 5 ml solution merits further discussion as our analysis showed no benefit of such an approach. 

Comment: In response to comment 8, The authors agreement is appreciated and thus are requested to merit with further discussion and highlight the same.

Comments 11: I would highly encourage the authors to try to modify the PLLA-SCA to actually understand the interaction or even propose potential strategies to improve the PLLA-SCA such that it can perform like it successful counter part. To do this i suggest trying to break the size distribution either by using an ultrasonic device or alternatively tweak the ion concentration of the solvent to shed some light on the actual behaviour and ways to influence the behaviour of this filler. Further testing, Zeta potential measurements recommended to gain insights into solution/suspension stability.

Response 11: The authors thank the reviewer for this comment and agree this would be an interesting line of research for a future study.

Comment: In response to comment 11, The authors are encouraged to include at least the simpler test that can be done in any wet lab. This will improve the impact of the paper and make it a worthy contribution.

Author Response

Thank you very much for taking the time to review this manuscript. Please find the detailed responses below and the corresponding revisions in the re-submitted files

Comments 5: For the observation in Figure 3 the authors are urged to detail the methodology, i.e., inspected volume / droplet-smear/ type of substrate etc. The figure is missing a scale bar and in general some formatting for better visual appeal.

Response 5: Figure 3 has in-figure labels. The smear was made from a drop with a volume of approximately 30 µm. The smear was covered with a 24 x 24 mm cover slip. There was no scale on the microscope.

Comment: In response to comment 5, The authors are requested to provide a scale bar, this can be done by imaging a known object (a ruler/ a stage micrometre etc.) and determining the magnification.

RESPONSE: A scale bar was uploaded to the microscope slides picture in the manuscript.

Comments 6: Discussion: How was  the CMC concentration detected and confirmed? Was it in accordance to the manufacture's description?

Response 6: The CMC concentration is in accordance with that outlined in the products’ respective instructions for use: PLLA-LASYNPRO (45 mg) and PLLA-SCA (90 mg).

Comment: In response to comment 6, the 'how' still remains unexplained.

RESPONSE: The CMC concentration was not calculated in this study. The statement in the discussion that ”PLLA-LASYNPRO™ formulation contains 50% less CMC than PLLA-SCA” was based on information obtained from the respective products’ instructions for use which note that PLLA-LASYNPRO contains 45 mg CMC/vial and PLLA-SCA contains 90 mg CMC per vial.

Comments 8: "Interestingly, the manufacturer's guidelines for PLLA-SCA (16) recommended a minimum reconstitution time of 120 minutes for more concentrated solutions; however, macroscopic analysis revealed no significant differences between solutions prepared with 5 mL or 8 mL of solvent, nor did reconstitution time appear to influence the overall macroscopic behavior of the product." - And why is that, what comments would the authors like to hypothesize?

Response 8: Based on observations comparing PLLA-SCA dissolved in 5 ml and 8 ml of solvent we believe the PLLA-SCA IFU recommendation for a reconstitution time of 120 minutes when using 5 ml solution merits further discussion as our analysis showed no benefit of such an approach.

Comment: In response to comment 8, The authors agreement is appreciated and thus are requested to merit with further discussion and highlight the same.

RESPONSE: Our findings are in agreement with those of Baumann et al, 2020 who reported no difference in PLLA-SCA particle shape or size distribution with standing times of 0, 2, 24 or 72 h. Shaking vigorously for 1 minute without a required standing time appears to have no effect on physicochemical properties. A solution prepared with 8 ml of solvent may help the withdrawal of product suspension from the vial without withdrawing foam.

Baumann K, et al. J Drugs Dermatol. 2020;19(12):1199-1203

Comments 11: I would highly encourage the authors to try to modify the PLLA-SCA to actually understand the interaction or even propose potential strategies to improve the PLLA-SCA such that it can perform like it successful counter part. To do this i suggest trying to break the size distribution either by using an ultrasonic device or alternatively tweak the ion concentration of the solvent to shed some light on the actual behaviour and ways to influence the behaviour of this filler. Further testing, Zeta potential measurements recommended to gain insights into solution/suspension stability.

Response 11: The authors thank the reviewer for this comment and agree this would be an interesting line of research for a future study.

Comment: In response to comment 11, The authors are encouraged to include at least the simpler test that can be done in any wet lab. This will improve the impact of the paper and make it a worthy contribution.

 RESPONSE: Searching for potential ways to improve distribution of PLLA-SCA is a natural conclusion from our study, but was not in our scope of interest for this research. It is important, however, and is the focus of a separate study from our laboratory. The data from this are in the validation stage and we do not want to publish any preliminary results at such an early stage. 

Round 3

Reviewer 2 Report

Comments and Suggestions for Authors

The authors are appreciated for trying to actively improve the manuscript. Of course, it is not always that all results can be put in one place. Nevertheless, the efforts put into improving overall value are worthy and i believe can move forward.